# Protection of K18-hACE2 Mice against SARS-CoV-2 Challenge by a Capsid Virus-like Particle-Based Vaccine

**DOI:** 10.3390/vaccines12070766

**Published:** 2024-07-12

**Authors:** Sebenzile K. Myeni, Anouk A. Leijs, Peter J. Bredenbeek, Shessy Torres Morales, Marissa E. Linger, Cyrielle Fougeroux, Sophie van Zanen-Gerhardt, Serge A. L. Zander, Adam F. Sander, Marjolein Kikkert

**Affiliations:** 1Molecular Virology Laboratory, Leiden University Center of Infectious Diseases (LU-CID), Leiden University Medical Center, 2333 ZA Leiden, The Netherlands; 2AdaptVac Aps, Ole Maaløes Vej 3, 2200 Copenhagen, Denmark; cfougeroux@adaptvac.com (C.F.);; 3Experimental Pathology Services Laboratory, Central Animal and Transgenic Facility, Leiden University Medical Center, 2333 ZA Leiden, The Netherlands; 4Experimental Animal Pathology Facility, The Netherlands Cancer Institute, 1066 CX Amsterdam, The Netherlands; 5Centre for Translational Medicine and Parasitology, Department for Immunology and Microbiology, Faculty of Health and Medical Sciences, University of Copenhagen, Blegdamsvej 3B, 2200 Copenhagen, Denmark

**Keywords:** SARS-CoV-2, cVLP-based COVID-19 vaccine candidate, ABNCoV2, RBD-cVLP, K18-hACE2 transgenic mice

## Abstract

The SARS-CoV-2 pandemic and the emergence of novel virus variants have had a dramatic impact on public health and the world economy, underscoring the need for detailed studies that explore the high efficacy of additional vaccines in animal models. In this study, we confirm the pathogenicity of the SARS-CoV-2/Leiden_008 isolate (GenBank accession number MT705206.1) in K18-hACE2 transgenic mice. Using this isolate, we show that a vaccine consisting of capsid virus-like particles (cVLPs) displaying the receptor-binding domain (RBD) of SARS-CoV-2 (Wuhan strain) induces strong neutralizing antibody responses and sterilizing immunity in K18-hACE2 mice. Furthermore, we demonstrate that vaccination with the RBD-cVLP vaccine protects mice from both a lethal infection and symptomatic disease. Our data also indicate that immunization significantly reduces inflammation and lung pathology associated with severe disease in mice. Additionally, we show that the survival of naïve animals significantly increases when sera from animals vaccinated with RBD-cVLP are passively transferred, prior to a lethal virus dose. Finally, the RBD-cVLP vaccine has a similar antigen composition to the clinical ABNCOV2 vaccine, which has shown non-inferiority to the Comirnaty mRNA vaccine in phase I-III trials. Therefore, our study provides evidence that this vaccine design is highly immunogenic and confers full protection against severe disease in mice.

## 1. Introduction

Severe acute respiratory syndrome coronavirus 2 (SARS-CoV-2), the causal agent of coronavirus disease 2019 (COVID-2019), was first reported in late 2019 in China [1], and continues to threaten public health and the global economy [2]. SARS-CoV-2 is highly transmissible between humans [3], and as of April 2024, there have been over 770 million confirmed cases of COVID-19 worldwide, with more than 7 million deaths, as reported by the World Health Organization (WHO). Multiple vaccines based on different platforms have been employed to control the COVID-19 pandemic [4]. However, the current arsenal of vaccines does not prevent (re-)infection or transmission [5] and thus allows for new variants of concern (VOCs) to evolve within the human population. These new virus variants are less sensitive to vaccine-elicited immunity, raising concerns for sustained vaccine efficacy [6]. The looming threat of future outbreaks because of SARS-CoV-2 evolution or related viruses and the logistical challenges associated with vaccine production and global distribution are driving the continued effort to develop additional scalable vaccine platforms that will mitigate transmission, provide sterilizing immunity, and are better scalable to ensure global vaccine equity [7].

VLPs are highly immunogenic due to their nanoparticle size and repetitive surface structure, and multiple successful vaccines based on VLPs have been licensed, including the human papillomavirus, hepatitis B virus, and hepatitis E virus vaccines [8]. To that end, the Tag/Catcher AP205 capsid virus-like particle (cVLP) vaccine platform [9] has previously been used to develop a novel COVID-19 vaccine, ABNCoV2, which employs cVLPs to display the receptor-binding domain (RBD) of the SARS-CoV-2 spike glycoprotein [10]. In a recent phase 1 clinical trial with healthy and naïve adults, the ABNCoV2 vaccine was well tolerated and induced strong SARS-CoV-2 neutralizing antibody responses [11]. Furthermore, in nonhuman primates (NHPs), the ABNCoV2 vaccine induced strong and long-lasting neutralizing antibody responses and efficacy against SARS-CoV-2 [12]. Finally, the unadjuvanted ABNCoV2 cVLP vaccine has been tested as a one-dose booster vaccine in a phase II and phase III study. In the latter study, ABNCoV2 showed a good safety profile and induced neutralizing antibody responses which had similar neutralizing capacity against the SARS-CoV-2 Wuhan strain to the licensed mRNA vaccine, Comirnaty [13,14].

Here, we evaluated for the first time the efficacy of an RBD-cVLP vaccine with a similar design to the clinical ABNCoV2 vaccine in K18-hACE2 transgenic mice. The K18-hACE2 mice express human ACE2 in the epithelium of respiratory airways and other organs, and represent a preclinical animal model of severe COVID-19 [15,16]. Preclinical studies of vaccines under development in animal models remain necessary not only to evaluate their immunogenicity but also to reflect the protective efficacy of COVID-19 vaccines [17]. Our studies confirm that SARS-CoV-2 infection and subsequent disease presentation are dose-dependent, and by using a prime–boost vaccination strategy, we show that the RBD-cVLP vaccine induces strong neutralizing antibody and cellular responses, which offer complete protection from virus-induced disease and mortality against a lethal SARS-CoV-2 challenge in the K18-hACE2 mouse model.

## 2. Materials and Methods

### 2.1. Cell Culture and Characterization of the SARS-CoV-2/Leiden-008 Isolate

VeroE6 cells were grown in Dulbecco’s modified Eagle’s medium (DMEM; Lonza, Verviers, France) supplemented with 8% fetal calf serum (FCS; Bodinco BV, Alkmaar, The Netherlands), 50 UI/mL penicillin (Lonza) and 50 µg/mL streptomycin (Lonza) at 37 °C and 5% CO_2_. The clinical isolate SARS-CoV-2/human/NLD/Leiden-008/2020 (SARS-CoV-2/Leiden-008) was isolated from a nasopharyngeal sample, and the NGS-derived sequence is available under GenBank accession number MT705206.1. The isolate differs from the original Wuhan strain and contains the D614G mutation in the spike protein and three additional non-silent mutations, namely C12846U in nsp9 (A54V), C14408U in nsp12 (P323L), and C18928U in nsp14 (P267S). Virus infections were performed in EMEM supplemented with 2% FCS, 50 IU/mL penicillin, and 50 µg/mL streptomycin. All experiments with live SARS-CoV-2 were performed in a biosafety level 3 (BSL-3) laboratory.

### 2.2. Animal Studies

#### 2.2.1. Ethics Declaration

All experiments involving mice were reviewed and approved by the Animal Experiments Committee of the LUMC and performed according to the recommendations and guidelines set by LUMC, the Dutch Experiments on Animals Act (DEC_20220310), and in strict accordance with EU regulations (2010/63/EU).

#### 2.2.2. Infection of K18-ACE2 Mice with SARS-CoV-2

Specific-pathogen-free, male, and female K18-hACE2 transgenic mice, which express the human ACE2 receptor under the control of the cytokeratin 18 (K18) promoter [15], were purchased from the Jackson Laboratory (B6.Cg-Tg(K18-ACE2)2Prlmn/J) and bred at the LUMC Central Animal Facility (PDC). All experiments involving SARS-CoV-2 were performed in the ABSL-3 unit of the LUMC Central Animal Facility (DM3). Mice were maintained in individually ventilated isolator cages (IsoCage Biocontainment System, Technniplast, Via l Maggio, Italy), provided sterile water and food ad libitum, and acclimated for seven days before the start of the experiment.

Male and female mice aged 8–12 weeks at the start of the experiment were pulled from several independent litters and randomized over experimental groups, and after transfer to the ABSL-3 facility, acclimated for a period of 7 days prior to the start of the experiments. To determine the lethal dose of the clinical isolate SARS-CoV-2/L-0008, a dosing experiment was performed. Mice were anesthetized with isoflurane and infected intranasally (i.n.) with dosages ranging from 0 to 1.25 × 10^5^ pfu/mouse diluted in DMEM to a volume of 50 µL. Mock (DMEM)-infected (controls) or intranasally (i.n.) infected mice with a specified viral dose of SARS-CoV-2 in a final volume of 50 µL were monitored daily for morbidity (body weight, disease symptoms) and mortality (survival). Mice showing >20% weight loss before the experimental set endpoints (2, 4, 6, or 14 days post infection) were defined as reaching their humane endpoints and humanely euthanized with an overdose of sodium pentobarbital (Euthasol 200 mg/kg). Mice that were also moribund were humanely killed at the discretion of the designated veterinarian and the researcher. At designated time points, macroscopic findings were noted for each mouse individually, and three samples were collected from each lung, one for determining virus titration/immune responses, one for RNA, and one for histology to score microscopic lesions.

#### 2.2.3. Vaccine and Immunizations

The RBD-cVLP vaccine was generated as previously described [10] and formulated using Addavax^TM^ (Invivogen, Toulouse, France). The RBD in the RBD-cVLP vaccine used in this study with a similar design to the clinical ABNCoV2 was designed with boundaries aa319–591 of the SARS-CoV-2 sequence (Sequence ID: QIA20044.1), which is homologous to both the isolate SARS-CoV-2/hu-77 man/NLD/Leiden-008/2020 (SARS-CoV-2/Leiden-008) and the original Wuhan strain. Male and female mice, aged 8–12 weeks, were randomized into three groups (group 1—PBS control; group 2—vehicle cVLPs; and group 3—RBD-cVLP). For immunizations, 2 µg of RBD-cVLP or vehicle cVLPs in 50 µL PBS were administered intramuscularly, in the thigh of one of the hind limbs under isoflurane anesthesia, using a two-week-interval prime–boost regimen. Blood sampling for serum isolation for binding and neutralizing antibody determination was collected on weeks 0, 2, and 4 via the tail-cut vein under isoflurane anesthesia. Spleens were collected 2 weeks after the second immunization to evaluate vaccine-induced cellular responses.

#### 2.2.4. Challenge of K18-hACE2 Mice

Four weeks after the first immunization, RBD-cVLP- and mock-immunized mice were challenged intranasally (i.n.) with a lethal dose of SARS-CoV-2 (10^4^ pfu in 50 µL DMEM) or mock-inoculated with DMEM. Morbidity/mortality status and weights were assessed and recorded daily for 14 days. On days 4 and 6, lungs were collected for virus titration by plaque assay and RT-PCR and for histopathology.

#### 2.2.5. Passive Serum Transfer

Pooled serum for passive transfer was obtained from K18-hACE2 transgenic mice that had been immunized (RBD-cVLP) or mock-immunized (control serum) as described above. The pooled sera from 10 mice were diluted 1:10 in PBS and administered intraperitoneally (i.p) at 250 µL. One day later, mice were challenged intranasally with a lethal dose (10^4^ pfu) of 50 µL of SARS-CoV-2. Daily health monitoring, survival, and body weights were recorded for 14 days as described above. Lungs were collected at 4 days post infection for virological analysis.

### 2.3. Techniques

#### 2.3.1. Lung Virus Titers

Viral titers were determined as previously described [18]. Briefly, lungs were weighed and homogenized using the gentleMACS dissociator by running the program Lung_02 (Miltenyi Biotec, Inc., Bergisch Gladbach, Germany) in 2 mL of PBS with 100 units/mL penicillin, 100 units/mL streptomycin (Lonza), 50 µg/mL gentamycin (Sigma-Aldrich, Amsterdam, Netherlands), and 0.25 µg/mL Fungizone (Gibco, Fisher Scientific, Landsmeer, The Netherlands). Tissue homogenates were pre-clarified at 300×*g* for 1 min and then further centrifuged at 10,000 RPM for 5 min, and supernatants were collected for measurement of viral load by plaque assay on VeroE6 cells and expressed as pfu/g lung.

The quantification of SARS-CoV-2 viral RNA was performed as previously described [18] using lung homogenates lysed with TriPure isolation reagent (Roche Applied Science, Woerden, The Netherlands) in gentleMACS M tubes (Miltenyi Biotec, Inc., Bergisch Gladbach, Germany) according to the manufacturer’s instructions. SARS-CoV-2 viral RNA was quantified by RT-qPCR using TaqMan Fast Virus 1-step mater mix (Thermo Fisher Scientific, Landsmeer, The Netherlands) on a CFX384 Touch Real-Time PCR Detection System (BioRad, Veenendaal, The Netherlands). The sub-genomic mRNA PCR primers and probes against the E gene and the genomic RdRp gene were modified based on previously described primer and probe sets [19]. The primers and probes were as follows: sub-genomic mRNA PCR primers (forward-GTGARATGGTCATGTGTGGCGG–RdRp_Sarbeco_F and reverse-CARATGTTAAASACACTATTAGCATA–RdRp_Sarbeco_R) and probe (FAM-CCAGGTGGAACMTCATCMGGWGATGC-BHQ1-RdRp_Sarbeco_Probe) and the genomic RNA PCR primers (forward-ACAGGTACGTTAATAGTTAATAGCGT–E_Sarbeco_F and reverse-ATATTGCAGCAGTACGCACACA–E_Sarbeco_R) and probe (TexRed-ACACTAGCCATCCTTACTGCGCTTCG-BHQ2). A standard curve was obtained using an in vitro transcript derived from a synthetic plasmid that contained all PCR targets. Each RNA sample was analyzed in triplicate.

#### 2.3.2. Histopathology and Semi-Quantitative Scoring of Lung Pathology

At necropsy, lungs were dissected from each mouse after euthanasia with an overdose of sodium pentobarbital, injected intraperitoneally under isoflurane anesthesia. The left lobe was then instillated with fixative (4% PFA in PBS) intratracheally, and immersion-fixed at room temperature for another 48 h. Then, samples were transferred into 70% ethanol and stored at 4 °C until further processing, which consisted of embedding in paraffin, cutting serial sections from the entire block at a nominal thickness of 5 µm, and mounting multiple sections per slide at a ratio of 1:10 for hematoxylin and eosin staining according to standard procedures. Stained slides were examined by an experienced veterinary pathologist, and microscopic findings were scored according to accepted principles [20] to calculate a semi-quantitative lung histopathology score for each animal, and an average score with standard deviation per experimental group.

#### 2.3.3. Neutralization Assay with an Authentic SARS-CoV-2 Strain D614G

Neutralization assays were performed in VeroE6 cells as previously described [18]. Briefly, heat-inactivated mouse sera were twofold serially diluted (starting from 1:10), mixed 1:1 with 120 TCID_50_/60 µL of SARS-CoV-2 in duplicates, and incubated for 1 h at 37 °C. Serum–virus mixtures were then added to Vero-E6 cell monolayers in 96-well plates and incubated at 37 °C for 3 days. Supernatants were then removed, and cells were fixed and virus inactivated with 40 µL 37% formaldehyde/PBS solution per well overnight at 4 °C. The fixative was removed from cells and the clusters stained with crystal violet solution. Cells were evaluated for cytopathic effect, and the virus neutralization titers were calculated as the reciprocal of the highest serum dilution that still inhibited virus replication. Each experiment included a SARS-CoV-2 back-titration to confirm that the dose of the used inoculum was within the acceptable range of 30 to 300 TCID_50_.

#### 2.3.4. ELISA Assay

ELISA assays were performed as previously described [18]. Briefly, lung homogenates used for virus titration as described above were used to evaluate the levels of interferons and cytokines. Homogenates were diluted 1:2 in diluent buffer and analyzed using ELISA kits from R&D Systems. ELISA kits specific for mouse IFN-β (DY8234-05, R&D Systems, Minneapolis, MN, USA), TNF-α (DY410-05, R&D Systems, Minneapolis, USA), IL-6 (DY406-05, R&D Systems, Minneapolis, MN, USA), IFN-λ (D485-05, R&D Systems, Minneapolis, MN, USA), IFN-_Υ_ (DY1789B-05, R&D Systems, Minneapolis, USA), and IL-1β (D401-05, R&D Systems, Minneapolis, USA) were used following the manufacturer’s specifications.

#### 2.3.5. IFN-γ ELISpot Assay

IFN-γ ELISpot was performed on mouse splenocytes isolated from vaccinated mice at 4 weeks post immunization using a mouse IFN-γ ELISpot-plus kit (Mabtech, Nacka Strand, Sweden) as previously described [18]. Briefly, spleens were mechanically dissociated through a sterile cell strainer and restimulated for 18–20 h at 37 °C with a pool of PepTivator SARS-CoV-2 Prot_S containing the sequence domains aa 304–338, 421–475, 492–519, 683–707, 741–770, 785–802, and 885–1273 (Catalog no. 130-126-701, Miltenyi Biotec, Leiden, The Netherlands) or aa 539–546 peptide pool at a final concentration of 1 μg/peptide/mL. The spike aa539–546 peptide (sequence: IKNQCVNFNFNGLTGTGVLTESNK) was produced at the peptide facility of the LUMC. Splenocytes were plated in triplicate wells, and the frequency of IFN-γ-secreting cells was determined using a mouse IFN-γ ELISpot-plus kit (Mabtech), following the manufacturer’s instructions. Spots in each well were quantified using an ELISpot reader and converted into number of spots per 1 million splenocytes for each well. The medium/unstimulated splenocytes were used as negative controls, and a CD3/CD28 mix (dilution 1:150) was used as a positive control. IFN-γ-secreting splenocytes were reported as the average of spot-forming cells (SFCs) per million splenocytes for each sample.

### 2.4. Statistical Analysis

Statistical significance was analyzed using GraphPad Prism software (GraphPad Software, version 9). Data were represented as mean ± SEM of at least 3 replicates. Statistical analysis was performed using Student’s *t*-test, or one-way analysis of variance (ANOVA) with Tukey’s post-test. The line represents the means and the error bar represents the SEMs. *p* values indicate significant differences (**** *p* < 0.0001, *** *p* < 0.001, ** *p* < 0.01 and * *p* < 0.05).

## 3. Results

### 3.1. K18-hACE2 Mice Develop a Dose-Dependent Disease When Challenged with the SARS-CoV-2/Leiden-008 Virus

The lethality and disease outcomes of the original SARS-CoV-2 Wuhan isolate in the K18-hACE2 mouse model have been described in several studies [16,21,22]. We confirmed the lethality of our own clinical isolate, SARS-CoV-2/Leiden-008 (described in the Section 2, and under GenBank accession number MT705206.1), in both female and male K18-hACE2 mice, aged 8–10 weeks (*n* = 10), in a dosing experiment using doses ranging from 0 to 1.25 × 10^5^ pfu per animal administered intranasally. Animals were monitored daily over a period of 14 days for clinical signs, body weight, and survival. Mice inoculated with either 2.5 × 10^4^ or 1.25 × 10^5^ pfu SARS-CoV-2 started losing weight at 2 days post infection, while mice inoculated with either 10^3^ or 5 × 10^3^ pfu SARS-CoV-2 started losing weight at 3 days post infection (Figure 1b), with drastically higher weight loss seen in the higher dosage groups. All infected animals irrespective of inoculation dose had the same disease outcomes, suffered significant weight loss, lethargy, ruffled fur, hunched posture, and irregular breathing. Animals reached their humane endpoints in a dose-dependent manner, with most animals in the higher dose groups reaching their human endpoints by day 5 or 6 post infection (Figure 1a). However, in the lowest dose group (10^3^ pfu), only 2 out 10 animals reached their humane endpoints 7 or 9 days post infection, respectively, while the other 8 animals regained their body weight and survived the infection (Figure 1a,b). All mock-infected animals showed no signs of morbidity and kept a stable body weight until the end of the experiment (Figure 1a,b). We next assessed the replication kinetics of SARS-CoV-2/Leiden-008 in lungs at days 2, 4, and 6 post infection at a lethal dose of 10^4^ pfu. Virus titers in the lungs were determined by plaque assay (Figure 1c) and viral RNA levels were determined by RT-qPCR (Appendix A). Regardless of the time after inoculation, there were no statistically significant differences in viral RNA levels between the lungs of mice at days 2, 4, and 6 post infection (Appendix A). Interestingly, the virus titer (infectious units) peaked at days 2 and 4 and significantly decreased at day 6 post infection (Figure 1c), indicating virus clearance over time.

Next, lungs from mice infected with SARS-CoV-2 or mock-infected with DMEM were examined microscopically on days 2, 4, and 6 post infection to investigate virus-induced lung pathology (Figure 1d,e). In contrast to mock-infected controls, in which only few interstitial macrophages and occasional neutrophils were observed in the alveolar septa (Figure 1d, top panels), SARS-CoV-2-infected animals invariably developed histiocytic interstitial pneumonia with a clear increase in extent and severity of the associated lesions over time post viral infection (Figure 1d, bottom panels). The SARS-CoV-2-induced pneumonia in the animals at day 6 post infection was characterized by multifocal to coalescing areas with more than 4-fold thickening of the alveolar septa compared with unaffected septa, infiltration by numerous macrophages, lymphocytes, and fewer viable neutrophils, and markedly increased numbers of macrophages and fewer neutrophils in the alveolar spaces. The perivascular space of small- to medium-sized arterioles was noticeably edematous, even at low (40×) magnification, and contained perivascular cuffs of mixed inflammatory cells dominated by lymphocytes and macrophages (Figure 1d,e, and Appendix A).

Taken together, these results demonstrate that K18-hACE2 mice are susceptible to the SARS-CoV-2/Leiden-008 isolate and that disease severity and lethality are dose-dependent. Furthermore, these results align with several independent studies [16,21,22], in which a similar disease phenotype in the lungs of this COVID-19 mouse model was observed.

### 3.2. Immunogenicity of the RBD-cVLP Vaccine in K18-hACE2 Transgenic Mice

To evaluate the immunogenicity of the RBD-cVLP vaccine, groups (*n* = 24) of 8–10-week-old K18-hACE2 mice were immunized intramuscularly with 2 µg RBD-cVLP or mock-vaccinated with PBS or 2 µg vehicle cVLPs at week 0 (prime) and week 14 (booster) (Figure 2a). Serum samples were collected at week 0 (before the prime vaccination; pre-sera), week 2 (before the booster), and week 4 (before SARS-CoV-2 challenge). RBD-cVLP-induced antigen-binding antibodies were determined 2 weeks following the second immunization, and all vaccinated mice developed RBD-specific IgG antibodies. No antigen-specific binding antibodies were determined in the mock-vaccinated groups, PBS, and vehicle cVLPs (Figure 2b). The neutralizing antibody responses against SARS-CoV-2/Leiden-008 were determined with a microneutralization assay as described in the Section 2. The neutralizing antibody titer is indicated as the dilution at which an inhibiting effect was visible and the cytopathic effect was fully prevented. All mice immunized with RBD-cVLP elicited robust neutralizing antibody titers to SARS-CoV-2 as early as 2 weeks post vaccination, and these further increased during the 2 weeks following booster immunization, reaching titers of up to 2560 (Figure 2c). In contrast, no virus neutralization was seen in the mock-vaccinated groups who received vehicle cVLPs or PBS (Figure 2c).

SARS-CoV-2-specific T-cell immunity is associated with reduced disease severity and can also influence antibody responses [23,24]. To assess S antigen-specific T cells by interferon-γ (IFN-γ) ELISPOT, we isolated splenocytes after prime–boost immunization and used S-specific peptide stimulation with two different peptide pools for activation upon in vitro culture. At 2 weeks following the booster vaccination, ELISPOT analysis revealed means of 290 IFN-γ SFCs in RBD-cVLP-vaccinated mice and 8 IFN-γ SFCs in control animals immunized with vehicle cVLPs (Figure 2d). Together, these results indicated that RBD-cVLP immunization induces both humoral and cellular immune responses in K18-hACE2 mice.

### 3.3. Protective Efficacy of RBD-cVLP in K18-hACE2 Mice Challenged with a Lethal Dose of SARS-CoV-2

We next evaluated the protective efficacy of RBD-cVLP in the stringent K18-hACE2 mouse model of severe COVID-19-like disease. For this purpose, RBD-cVLP prime–boost-immunized or mock-immunized (vehicle/cVLPs) mice were intranasally challenged with a lethal dose (10^4^ pfu) of SARS-CoV-2 and monitored daily for weight loss, survival, and viral replication (Figure 2a). Interestingly, all animals (12/12) that were immunized with RBD-cVLP and challenged with a lethal dose of SARS-CoV-2 survived until the end of the experiment (Figure 3a). However, 10 out of 12 mock-immunized mice (vehicle/cVLPs) challenged with a lethal dose of SARS-CoV-2 succumbed to infection before or on day 9 (Figure 3a). Additionally, while most of the SARS-CoV-2-challenged mock-vaccinated mice started to lose body weight rapidly from day 3 post viral challenge, no body weight loss or ill health was observed in the RBD-cVLP-vaccinated mice (Figure 3b). At days 4 and 6 post challenge, the viral loads in the lungs of RBD-cVLP-vaccinated mice were below the lower limit of detection (10 pfu/mL). In contrast, in mock-immunized (vehicle/cVLPs) mice, viral loads were on average as high as 3.46 × 10^6^ pfu/g/lung at day 4 and 3.29 × 10^5^ pfu/g/lung at day 6 (Figure 3c). Significantly high levels of both genomic and sub-genomic viral RNA copies were detected in the lungs of mock-immunized (vehicle/cVLPs) mice, consistent with the high levels of infectious virus units observed at days 4 and 6 post virus challenge. In contrast, the RBD-cVLP-vaccinated mice had significantly reduced viral RNA loads (at least 3-log reduction) in their lungs at day 4 and no detectable viral RNA levels at 6 days post challenge (Figure 3c and Appendix A).

Microscopic examination of the lungs from mock-vaccinated controls confirmed the results of the initial infection experiment with the SARS-CoV-2/Leiden 008 isolate, as animals also developed (lympho)histiocytic interstitial pneumonia with similar characteristics over time post infection as described above (albeit with some variation in extent and severity of the associated lesions and between animals; Figure 3d and Appendix A). However, in the RBD-cVLP-vaccinated animals, at day 4 post challenge, a trend towards lower combined lung pathology scores was observed, and at day 6, this difference with the mock-vaccinated controls was statistically highly significant (*p* < 0.001; Figure 3e). This was due to a noticeable increase in combined lung pathology scores in the mock-vaccinated animals on day 6 post challenge, whereas scores in RBD-cVLP-vaccinated animals remained on similarly modest levels to those on day 4, while variability in scores decreased for both groups from days 4 to 6 post challenge.

Taken together, these results demonstrate that immunization with RBD-cVLP prior to a lethal SARS-CoV-2 challenge completely protects against death and is associated with only mild lung pathology without noticeable signs of clinical disease.

### 3.4. RBD-cVLP Vaccination Reduces Lung Inflammatory Cytokine Responses to a Lethal Challenge with SARS-CoV-2

The elevation of cytokine and chemokine concentrations during the acute phase of COVID-19 has been associated with an increased risk of disease severity and mortality [25]. Since mice vaccinated with RBD-cVLP demonstrated reduced lung pathology and were protected from a lethal dose of SARS-CoV-2, we evaluated the levels of IL-1β, IL-6, TNF-α, IFN-γ, and IFN-λ in lung homogenates of RBD-cVLP- and vehicle/cVLP-vaccinated mice at days 4 and 6 post challenge with SARS-CoV-2. Compared with the mice from the vehicle/cVLP (mock)-vaccinated group, the lungs from mice that were vaccinated with RBD-cVLP showed a significant reduction in the levels of IL-1β, IL-6, TNF-α, and IFN-γ at both days 4 and 6 (Figure 4). In contrast, no differences in levels of IFN-λ were observed in the RBD-cVLP-vaccinated and vehicle/cVLP-vaccinated lungs (Figure 4). Collectively, these results indicated that vaccination with the prime and boost dose of 2 µg of RBD-cVLP reduced the induction of proinflammatory cytokines by SARS-CoV-2.

### 3.5. Passive Transfer of Immune Sera from RBD-cVLP-Vaccinated Mice Increases the Survival of Naïve K18-hACE2 Mice from a Lethal SARS-CoV-2 Infection

To investigate the protective efficacy of antibodies induced by immunization with RBD-cVLP, a passive transfer experiment was carried out by intraperitoneal inoculation of pooled sera from mice vaccinated intramuscularly with RBD-cVLP or mock-vaccinated with vehicle/cVLPs into naïve K18-hACE2 mice. One day after serum transfer, 8–10-week-old naïve K18-hACE2 mice were challenged via an intranasal route with a lethal dose of 10^4^ pfu of SARS-CoV-2. Both groups of mice that received sera from either RBD-cVLP- or vehicle/cVLP (mock)-vaccinated mice initially lost weight; however, the mock-treated mice that received sera from vehicle/cVLP-induced antibodies lost significantly more weight, and 8 out of 10 animals succumbed to disease by 7 days post challenge. In contrast, most mice (7/10) that received neutralizing antibody serum rapidly regained weight in the ensuing period and survived until the end of the experiment (Figure 5a,b). On day 4 post virus challenge, viral lung titers in all the animals that received serum from RBD-cVLP-immunized mice were significantly lower compared to animals that had received mock serum (Figure 5c). These data confirmed that the virus-neutralizing antibodies induced by vaccination with RBD-cVLP play a key role in protecting K18-hACE2 mice from a lethal challenge with SARS-CoV-2.

## 4. Discussion

While the SARS-CoV-2 virus outbreak has reached an endemic stage, new variants of concern (VOCs) continue to arise and spread. Vaccine protection with the current vaccines against SARS-CoV-2 infection quickly wanes over time, requiring regular updated boosters [5,26,27]. Thus, continuous efforts to monitor the spread of the SARS-CoV-2 virus and the development of new vaccines that are efficacious, easy to disseminate and provide long-lasting and sterilizing immunity remain a high priority. In this study, we evaluated in mice the protective efficacy of a second-generation RBD-cVLP vaccine against SARS-CoV-2.

We first evaluated the ability of our own clinical isolate SARS-CoV-2/L-0008 to infect K18-hACE2 mice, which expresses hACE2 under the K18 epithelial promotor [15]. Similar to the original Wuhan strain, our clinical isolate, which also contains the D614G mutation in the spike protein [28], showed a dose-dependent increase in morbidity and mortality over nine days, corroborating other studies with the K18-hACE2 mice [16,21,29,30,31,32,33], which exhibit a fatal respiratory infection following intranasal administration of SARS-CoV-2. Infection with a lethal SARS-CoV-2/L-0008 dose induced histiocytic interstitial pneumonia with a clear response pattern and increased severity over time, including histological aspects of ARDS seen in COVID-19 patients. The dose-dependent sub-lethal disease manifestation and the lung histopathological findings of COVID-19-associated ARDS makes this model highly suitable for testing SARS-CoV-2 countermeasures including vaccines.

Immunization of C57BL/6 K18-hACE2 mice with RBD-cVLP induced robust binding and neutralizing antibodies against the historical SARS-CoV-2 Leiden-008/2020 D614G, with similar efficacy seen to sera from RBD-cVLP-vaccinated BALBc mice [10], which are known to produce a stronger humoral response than C57BL/6 mice [34,35]. Vaccination with RBD-cVLP conferred complete protection against a lethal SARS-CoV-2 challenge in K18-hACE2 mice, providing virtually complete virological protection with no infectious virus recovered from the lungs of RBD-cVLP-vaccinated mice at days 4 and 6 post challenge. More than 66% of the RBD-cVLP-vaccinated animals had no detectable amounts of sub-genomic E and genomic RdRp mRNAs in the lungs, while only ~33% of the animals at day 4 after challenge had significantly reduced viral RNA, indicating that abortive SARS-CoV-2 replication occurred. Importantly, no viral RNA was detected in RBD-cVLP-vaccinated mice at day 6 post challenge, indicating complete viral RNA clearance by day 6 in all RBD-cVLP-vaccinated animals, which is further supported by the lack of clinical symptoms and weight loss, reduced inflammation and the limited lung pathology observed at day 6 post challenge in a stringent and lethal model.

It is becoming clear that the ideal vaccine against COVID-19 disease would need to generate high levels of cross-neutralizing antibodies, Fc effector antibodies, and T cells (CD4+ and CD8+) against the variant viruses that are sustained for a long time [36,37]. In addition, SARS-CoV-2 vaccines with enhanced mucosal immunity may be required to reduce break-through infections and subsequent transmission [38]. At this time, we lack such a vaccine, and both humoral and cellular responses induced by currently approved SARS-CoV-2 vaccines wane over time, resulting in a need for booster vaccinations [26]. Particulate antigens including both live-attenuated viruses and VLPs [39] remain the only vaccine immunogens that have proven capable of inducing durable antibody responses, even after a single dose. Thus, there is a clear rationale for delivering the SARS-CoV-2 RBD antigen in a similar particulate format.

Passive transfer of serum from RBD-cVLP-vaccinated mice suggested that antibodies play a significant role in the protection of K18-hACE2 mice against a fatal SARS-CoV-2 infection by decreasing viral loads in the lungs and also providing a substantial survival advantage. We also observed that RBD-cVLP immunization induces SARS-CoV-2-specific T-cell responses in mice, suggesting that both RBD-cVLP-stimulated robust neutralizing antibodies and T-cell responses are required for sufficient protection. One of the benefits of the modular Tag-Catcher-AP205 capsid-like particle vaccine design of RBD-cVLP is the flexibility in quickly replacing the vaccine antigen when any new SARS-CoV-2 variant arises. cVLPs are also highly immunogenic due to their size and repetitive surface epitope display and provide an alternative as the next-generation COVID-19 booster vaccine candidate. The protein-based RBD-cVLP vaccine is not expected to require ultra-cold storage conditions (–20 and –70 °C), as opposed to currently approved mRNA-based COVID-19 vaccines, and can also be freeze-dried without loss of efficacy [40], easing global distribution. 

In conclusion, the RBD-cVLP vaccine induces potent neutralizing antibodies and can protect against severe SARS-CoV-2 infection and lung disease in a fatal mouse challenge model. These findings support the continued development of RBD-cVLP-based vaccines for managing COVID-19 and future SARS-CoV-2 outbreaks. 

## Figures and Tables

**Figure 1 vaccines-12-00766-f001:**
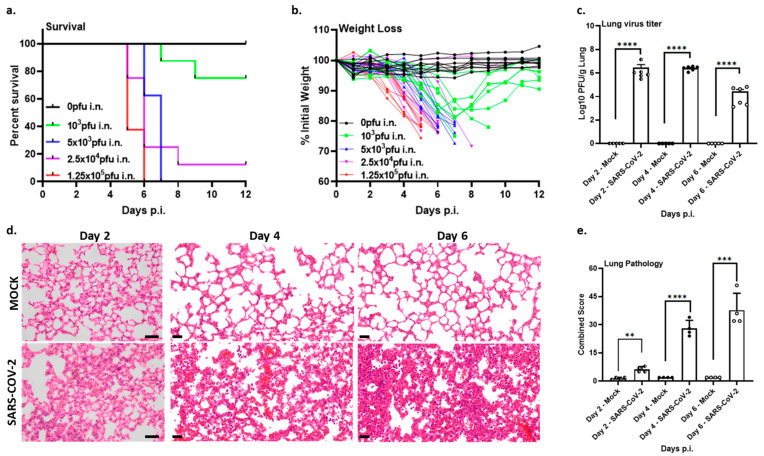
K18-hACE2 mice develop a dose-dependent disease when challenged with the SARS-CoV-2/Leiden-008 virus. Groups of 10 or 6 mice were infected with indicated doses of SARS-CoV-2 ranging from 10^3^ to 1.25 × 10^5^ pfu or mock-inoculated with DMEM. (**a**) Survival percentages were monitored for 14 days (*n* = 10). (**b**) Relative body weight loss (% from initial weight) (*n* = 10). (**c**) Virus titers in lungs at days 2, 4, and 6 post inoculation. Groups of 6 mice were infected intranasally with SARS-CoV-2 (10^4^ pfu) or mock-infected (DMEM). Titers are shown in pfu per gram of lung tissue (PFU/g lung) and symbols represent each individual mouse, half open circles represent day 2, closed circles represent day 4, and open circles represent day 6 mice. The limit of detection for infectious virus is 10 pfu/g lung. (**d**) Lung pathology at days 2, 4, and 6 post inoculation. Photomicrographs with representative lesions from mock- (top row of panels) and SARS-CoV-2 infected animals (bottom row of panels, H&E stain, scale bars on the bottom right or left corner, 20 µm, 40 ×) are shown. (**e**) Bar chart with semi-quantitative combined lung pathology scores. symbols represent each individual mouse, half open circles represent day 2, closed circles represent day 4, and open circles represent day 6 mice. Bar heights indicate group means (*n* = 4) and error bars the standard error of the mean. ns: not significant, ** *p* < 0.01, *** *p* < 0.001, **** *p* < 0.0001.

**Figure 2 vaccines-12-00766-f002:**
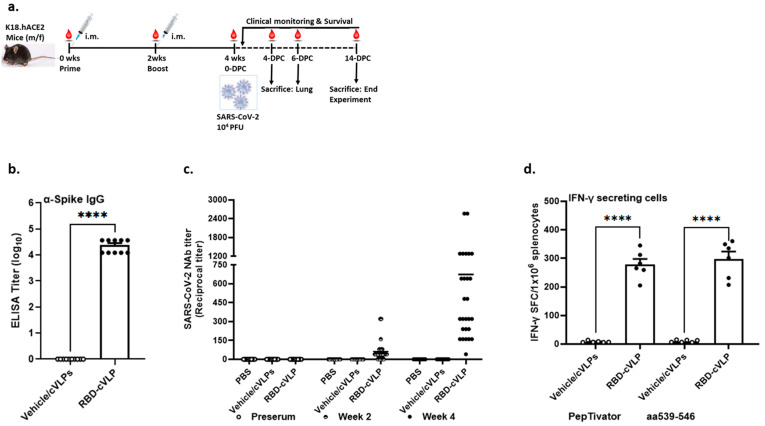
Immunogenicity of the RBD-cVLP vaccine in K18-hACE2 mice. (**a**) Scheme of immunizations, blood collection, and virus challenge. Groups of 24 mice were immunized intramuscularly with 2 µg of RBD-cVLP or mock-immunized with vehicle cVLPs or PBS in 50 µL; sera were collected at 0, 2, and 4 weeks after first immunization. (**b**) Serum spike-specific IgG levels were determined at 4 weeks after first immunization. (**c**) Serum neutralizing antibody titers were determined against the SARS-CoV-2 at indicated times post immunization. (**d**) SARS-CoV-2-specific T-cell immune responses. Magnitude of SARS-CoV-2-specific cell responses directed against the spike. IFN-γ was evaluated by an ELISpot assay from splenocytes derived from immunized mouse groups at 4 weeks post vaccination. Bars show mean values and the standard deviation. ns: not significant, **** *p* < 0.0001.

**Figure 3 vaccines-12-00766-f003:**
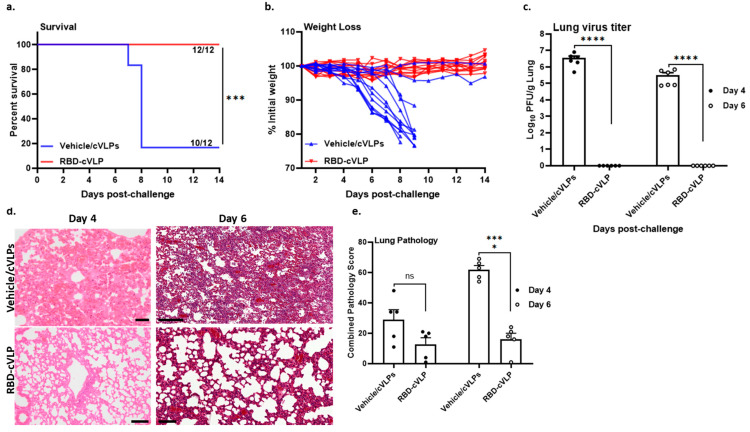
Protection against a lethal dose of SARS-CoV-2 infection after RBD-cVLP vaccination in K18-hACE2 mice. RBD-cVLP prime–boost-immunized or mock-immunized (vehicle/cVLPs) mice were intranasally challenged with a lethal dose (10^4^ pfu) of SARS-CoV-2. (**a**) Survival percentages were monitored for 14 days (*n* = 12). (**b**) Relative body weight loss (% from initial weight) (*n* = 12). (**c**) Virus titers in lungs at days 4 and 6 post challenge. Groups of 6 RBD-cVLP- or mock-immunized (vehicle/cVLPs) mice were challenged intranasally with SARS-CoV-2 (10^4^ pfu). Titers are shown in pfu per gram of lung tissue (PFU/g lung) and symbols represent each individual mouse. The limit of detection for infectious virus is 10 pfu/g lung. (**d**) Lung pathology at day 4 and 6 post challenge (*n* = 5). Lungs of all animals were examined microscopically, and photomicrographs with representative lesions from mock- (top row of panels) and RBD-cVLP-immunized animals (bottom row of panels, H&E stain, scale bars on the bottom right or left corner, H&E stain, scale bars on the bottom right or left corner, Vehicle/cVLPs:Day 4–50 µm; Vehicle/cVLPs:Day 6–200 µm; RBD-cVLP:Day 4–20 µm; RBD-cVLP:Day 6–100 µm 40 ×) are shown. (**e**) Bar chart with semi-quantitative combined lung pathology scores. Bar heights indicate group means (*n* = 5) and error bars the standard error of the mean. ns: not significant, * *p* < 0.05, *** *p* < 0.001, **** *p* < 0.0001.

**Figure 4 vaccines-12-00766-f004:**
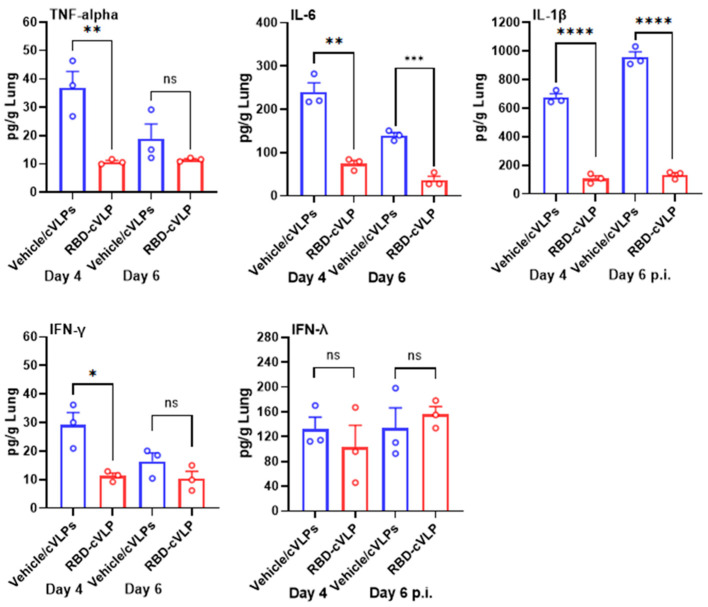
RBD-cVLP vaccination reduces lung inflammatory cytokine responses to a lethal challenge with SARS-CoV-2. TNFα, IL-6, IL-1β, IFN-ϒ, and IFN-λ protein levels were quantified by ELISA using the same lung homogenates used for measuring viral titers (see Figure 3) extracted from RBD-cVLP-vaccinated and unvaccinated K18-hACE2 mice at days 2 and 4 post challenge with SARS-CoV-2. Data are represented as means (pg/g lung) ± SEM. ns: not significant, * *p* < 0.05, ** *p* < 0.01, *** *p* < 0.001, **** *p* < 0.0001.

**Figure 5 vaccines-12-00766-f005:**
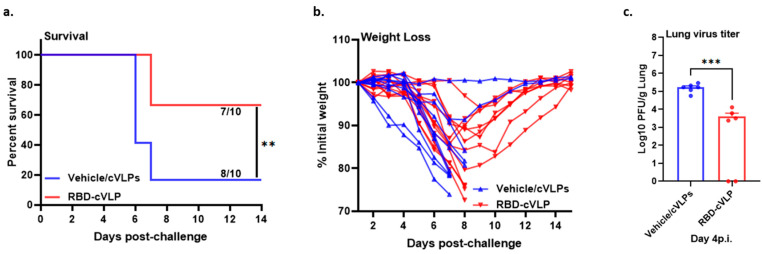
Challenge of K18-hACE2 mice after passive transfer of immune sera from mice immunized with RBD-cVLP. Naïve K18-hACE2 mice were injected intraperitoneally with 250 µL of pooled serum from K18-hACE2 mice immunized with 2 µg of RBD-cVLP or mock-immunized with vehicle cVLPs. At 24 h post serum transfer, mice were intranasally challenged with a lethal dose 10^4^ pfu of SARS-CoV-2. (**a**) Survival percentages were monitored for 14 days (*n* = 10). (**b**) Relative body weight loss (% from initial weight) (*n* = 10). (**c**) Virus titers in lungs at days 4 post challenge (*n* = 6). Lungs were harvested from mice at day 4 post SARS-CoV-2 challenge then homogenized, and virus titers were measured by plaque assay on VeroE6 cells. The mean ± SEM per group and virus titer in pfu per gram of lung tissue are presented. Symbols represent individual mice. The limit of detection for infectious viral progeny is 10 pfu/g lung. ns: not significant, ** *p* < 0.01, *** *p* < 0.001.

## Data Availability

All data supporting the reported results can be found in the published paper.

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
