# Peer review of "Protection of K18-hACE2 Mice against SARS-CoV-2 Challenge by a Capsid Virus-like Particle-Based Vaccine"

_vaccines, 2024, doi:10.3390/vaccines12070766_

Round 1

Reviewer 1 Report

Comments and Suggestions for Authors

The paper by Myeni et al clearly demonstrates that their VLP-based vaccine against CSARS-CoV--2 protects ACE2-transgenic mice. While the data are fine, they try too much to hide that their vaccine is a blow from the past dealing with the Wuhan strain. In addition, the vaccine is not even described in materials section or elsewhere. No attempt is made to make an Omicron vaccine based on the same strategy (we failed). The most glaringly misleading statement, which actually comes out of a press release from Bavarian Nordic is the following: Our vaccine is non-inferior compared a current RNA vaccine. Only reading the press-release and looks up what actually the RNA vaccine is, you find out that they compare their Wuhan-based vaccine with a pure Omicron vaccine and test the neutralising antibodies on Wuhan. Not Omicron. This statement in conjunction with a phase III press release of a listed company is obviously unethical and questionably legal. Hence, the whole paper is well done and technically sound. But a blow from the past misleadingly packaged in something looking relevant.

Author Response

We thank the reviewer for his/her appreciation that the data is fine and the whole paper is well done and technically sound. We are happy to provide further clarifications to address the reviewer’s concerns as outlined below.

  • During the revision process we also realized that we need to make it clear that the tested vaccine is not It is correct that it has a similar design and antigen composition - but in order to say ABNCOV2, we should have used the clinical batch - which is not the case in this study. We apologize for the oversight. We have now corrected this throughout the manuscript, figures and made some few other changes to the text (highlighted throughout the text) to provide further clarity. Thus all our arguments/discussion points for why it is interesting to test this vaccine still holds.
  • The RBD-cVLP vaccine used in this study has been previously described before and cited in the Methods section (line 130), reference 10 (line 591-2). Fougeroux, C., et al., Capsid-like particles decorated with the SARS-CoV-2 receptor-binding domain elicit strong virus neutralization activity. Nat Commun, 2021. 12(1): p. 324. That is why the vaccine was not extensively described.
  • The goal of this study was to evaluate the protective efficacy of the RBD-cVLP vaccine based on the Wuhan strain in a stringent/sensitive lethal challenge mouse model. We provide evidence that the RBD-cVLP vaccine is highly immunogenic and confers full protection against a severe disease in mice supporting continued development of RBD-cVLP based vaccines. This is a proof of principle study in an animal model and the design of an Omicron or other variants of SARS-CoV-2 vaccine based on the same strategy is beyond the scope of this work.
  • To address the comment made about press release from Bavarian Nordic and the phase III press release; in the phase III study the ABNCOV2 vaccine was benchmarked against mRNA corminaty vaccine. Both vaccines are based on Wuhan. Thus the above comment is not right and our goal is not to communicate misleading information or statement.

Reviewer 2 Report

Comments and Suggestions for Authors

Protection of K18-hACE2 mice against SARS-CoV-2 challenge by a Capsid Virus-Like Particle-Based vaccine.

vaccines-3041889

In this study the authors evaluated the pathogenicity of the SARS-CoV-2/Leiden_008 isolate in K18-hACE2 transgenic mice. Using this isolate the authors shoed that the capsid virus-like particle (cVLP)- based vaccine, ABNCoV2, induces strong neutralizing antibody responses and sterilizing immunity in mice. Additionally, the authors aldo demonstrated that vaccination with ABNCoV2 can also protect mice  from  the lethal infection  disease. The authors also showed that the survival of naïve animals significantly increases when sera from mice vaccinated with ABNCoV2 are passively transferred, prior to a lethal virus dose. 

Overall the the research study is perfectly described and presented by the authors. All the sections including introduction, methods, results and conclusion have been described perfectly. 

The manuscript can be accepted in the present form. 

Author Response

We thank the reviewer for his/her positive words on our manuscript and for his/her appreciation for the quality of our manuscript. During the revision process we also realized that we need to make it clear that the tested vaccine is not ABNCOV2.  It is correct that it has a similar design and antigen composition - but in order to say ABNCOV2, we should have used the clinical batch - which is not the case in this study. We apologize for the oversight. We have now corrected this throughout the manuscript, figures and made some few other changes to the text (highlighted throughout the text) to provide further clarity. Thus all our arguments/discussion points for why it is interesting to test this vaccine still holds.

Reviewer 3 Report

Comments and Suggestions for Authors

In this study, the authors showed that the capsid virus-like particle (cVLP)- based vaccine, ABNCoV2, induces strong neutralizing antibody responses and sterilizing immunity in K18-hACE2 mice and protects mice from both a lethal infection and symptomatic disease of the SARS-CoV-2/Leiden_008 isolate. 

Several suggestions:

1.      Is the RBD in the ABNCoV2 (in line 52) homologous to the isolate SARS-CoV-2/hu-77 man/NLD/Leiden-008/2020 (SARS-CoV-2/Leiden-008) and/or original Wuhan strain?

2.      2.2.1. Ethics Studies. It is better to have the certificate number.

3.      Line 94, please change reference [ (McCray et al, 2007)] to [14].

4.      Line 164, no [(Corman 164 et al, 2020)] in the reference list.

5.      Line 200, [Wild-type] means Wuhan strain, not SARS-CoV-2/hu-77 man/NLD/Leiden-008/2020 (SARS-CoV-2/Leiden-008)?

6.      Line 218 and line 226, same [Enzyme-linked immunosorbent assay (ELISA)]. Combines together?

7.      Please cite the reference(s) for PepTivator SARS-CoV-2 and aa 539-549 in lines 230-232 to mention these peptides are epitopes or immunogens.

8.      Both [spike] and [Spike] are used in the article. Please unify.

9.      Figure 4, [pg/mL] was used in the figure while [pg/g lung] in figure legend. Please unify.

Author Response

We thank the reviewer for his/her statement and clear understanding of the work that is presented in this manuscript. During the revision process we also realized that we need to make it clear that the tested vaccine is not ABNCOV2.  It is correct that it has a similar design and antigen composition - but in order to say ABNCOV2, we should have used the clinical batch - which is not the case in this study. We apologize for the oversight. We have now corrected this throughout the manuscript, figures and made some few other changes to the text (highlighted) to provide further clarity.

We are happy to provide further clarifications to the reviewer’s suggestions as outlined below.

Several suggestions:

  1. Is the RBD in the ABNCoV2 (in line 52) homologous to the isolate SARS-CoV-2/hu-77 man/NLD/Leiden-008/2020 (SARS-CoV-2/Leiden-008) and/or original Wuhan strain?

The RBD in the RBD-cVLP vaccine used in this study with a similar design as the clinical ABNCoV2 was designed with boundaries aa319-591 of the SARS-CoV-2 sequence (Sequence ID: QIA20044.1), which is homologous to both the isolate SARS-CoV-2/hu-77 man/NLD/Leiden-008/2020 (SARS-CoV-2/Leiden-008) and original Wuhan strain. The isolate SARS-CoV-2/Leiden-008 available under GenBank accession number MT705206.1 differs from the original Wuhan strain and contains the D614G mutation in the spike protein, which lies outside the RBD. The description on how the RBD-cVLP vaccine was generated is also cited in the Methods section (line 130). We have now added this information in the Methods section (line 131-135) to provide more clarity to the reader.

  1. 2.2.1. Ethics Studies. It is better to have the certificate number.

We thank the reviewer for noting this and have added the certificate number, DEC_20220310 (line 99).

  1. Line 94, please change reference [ (McCray et al, 2007)] to [14].

We thank the reviewer for noting this and have changed this (line 103).

  1. Line 164, no [(Corman 164 et al, 2020)] in the reference list.

We thank the reviewer for noting this and have changed this (line 177)

  1. Line 200, [Wild-type] means Wuhan strain, not SARS-CoV-2/hu-77 man/NLD/Leiden-008/2020 (SARS-CoV-2/Leiden-008)?

Wild type means the SARS-CoV-2/Leiden-008 (GenBank accession number MT705206.1) throughout the manuscript. We did not use the Wuhan strain in this study. We have now changed this to “Neutralization assay with an authentic SARS-CoV-2 strain D614G” (line 213-214).

  1. Line 218 and line 226, same [Enzyme-linked immunosorbent assay (ELISA)]. Combines together?

We thank the reviewer for noting this and have corrected this to make it clearer. Line 231 is “ELISA assay” and line 239 is “IFN-γ ELISpot assay.

  1. Please cite the reference(s) for PepTivator SARS-CoV-2 and aa 539-549 in lines 230-232 to mention these peptides are epitopes or immunogens.

The pool of PepTivator SARS-CoV-2 Prot_S containing the sequence domains aa 304-338, 421-475, 492-519, 683-707, 741-770, 785-802, and 885 – 1273 was purchased from Miltenyi Biotec, which is already stated in line 246. We have now added the catalog number. The Spike aa539-546- peptide (sequence: IKNQCVNFNFNGLTGTGVLTESNK) was produced at the peptide facility of the LUMC. The purity of the synthesized peptide (75–90%) was determined by HPLC and the molecular weight by mass spectrometry. We have now added the additional information on the aa539-546 peptide (line 247-250).

  1. Both [spike] and [Spike] are used in the article. Please unify.

We thank the reviewer for noting this and have now used “spike” throughout the article. Please see line 59, 90, 246, 351, 355, 486 and 620.

  1. Figure 4, [pg/mL] was used in the figure while [pg/g lung] in figure legend. Please unify.

We thank the reviewer for noting this and have now changed the y-axis title in Figure 4 to pg/g Lung. The total protein concentrations (pg/mL) were normalized to the lung weight of each mouse.

Round 2

Reviewer 1 Report

Comments and Suggestions for Authors

The paper is much improved now. What is still missing, however, is a comparison of their particular VLP-based construct with other VLP based constructs.

Author Response

Reviewer 1

Comments and Suggestions for Authors

The paper is much improved now. What is still missing, however, is a comparison of their VLP-based construct with other VLP based constructs.

We thank the reviewer for his/her positive evaluation of the revised manuscript and we appreciate the valuable suggestion.

We agree with the reviewer that ideal future experiments (beyond the scope of this study) will have to look into how our VLP-based construct compares with other VLP based constructs in same experimental setting. In line 52-66, we mention the potential advantages of using VLPs (highly immunogenic) and we have also demonstrated in our manuscript that our modular Tag-Catcher-AP205 capsid-like design of RBD cVLP is highly immunogenic and protective against a sensitive and lethal mouse model resulting in undetectable infectious virus units after challenge.